# Can Low-Iodine, Low-Radiation-Dose CT Aortogram Reliably Detect Endoleak after Endovascular Aneurysm Repair of the Aorta?

**DOI:** 10.3390/diagnostics13132228

**Published:** 2023-06-30

**Authors:** Po-An Chen, Eric P. Huang, Yi-Chun Chen, Chiung-Chen Chuo, Shu-Tin Huang, Ming-Ting Wu

**Affiliations:** 1Department of Radiology, Kaohsiung Veterans General Hospital, No. 386, Ta-Chung 1st Road, Kaohsiung 813, Taiwan; evaporativem@gmail.com (P.-A.C.); jumpjump612@gmail.com (Y.-C.C.);; 2Faculty of Medicine, School of Medicine, National Yang Ming Chiao Tung University, Taipei 112, Taiwan; erich0600@gmail.com; 3Department of Radiology, Park One International Hospital, No. 100, Bo’ai 2nd Road, Kaohsiung 813, Taiwan; 4Department of Radiology, New Taipei City Hospital, No. 3, Sec. 1, New Taipei Blvd., Sanchong Dist., New Taipei City 241, Taiwan; 5Institute of Clinical Medicine, National Yang Ming Chiao Tung University, Taipei 112, Taiwan

**Keywords:** double-low computed tomography angiography, lower tube voltage, low-contrast media, endovascular aneurysm repair, aortic computed tomography angiography, radiation dosage

## Abstract

Objective: Double-low CT aortography (DLCTA) is increasingly used in follow-up studies of aortic aneurysm after endovascular aneurysm repair (EVAR). However, whether DLCTA can reliably detect the presence of endoleak is not clear. Methods: From February 2014 to October 2019, patients who received EVAR, underwent CT surveillance, and had at least one standard CTA protocol (120 kVp, 400 mg I/kg) and one DLCTA (70–80 kVp, 200 mg I/kg) were included. The integrated findings of the standard CTA and sequential change were considered as the reference standard for the presence of endoleak. Results: In all, 36 patients received TEVAR and 24 patients received EVAR; 62 standard CTA and 167 DLCTA results were analyzed. There were 2 type I (3.3%) and 12 type II (20.0%) endoleaks in 14 patients (23.3%). The performance of DLCTA in the diagnosis of endoleak reached 100% accuracy compared to that of standard CTA in case of the correction of CT findings by an expert second reading. Compared to the standard CTA, DLCTA scan reduced the radiation dose by 71% and the iodine dose by 50%. Conclusions: DLCTA with 70–80 kVp and 200 mg I/kg can reliably detect the presence of endoleak after TEVAR/EVAR.

## 1. Introduction

Endovascular aneurysm repair (EVAR) is the primary treatment for abdominal and thoracic aortic aneurysms as an alternative to open surgery [1,2,3,4]. Endoleak development is defined as the persistence of blood flow outside the stent graft and within the aneurysm after EVAR. It represents a significant risk factor for the growth and subsequent rupture of the aneurysm sac [5,6,7]. Therefore, to ensure long-term success after EVAR and prevent late rupture, which may be asymptomatic and potentially fatal, lifelong interval surveillance imaging is considered essential [3,4,8].

Computed tomography angiography (CTA) is the standard imaging tool for serial pre- and postoperative evaluation of an aortic aneurysm [3,4]. However, the long-range scanning coverage and typically triple phases of the whole aorta result in a high radiation exposure [9]. Furthermore, the usage of large amounts of contrast medium (CM) causing contrast-induced nephropathy is an important safety issue in CTA, especially since most EVAR patients are elderly and have compromised renal function [10,11]. Previous reports revealed that approximately one-third of patients with aortic aneurysms undergoing surgery or endovascular treatment had chronic kidney disease [12].

Double-low CT aortography (DLCTA) of the aorta, a protocol with low tube voltage and a low amount of contrast agent, is increasingly used in aortic CTA. The tube voltage ranges from 70 to 100 kVp, with an iodine load of 200–300 mgI/kg, as in previous studies [13,14,15,16,17,18]. An earlier report showed that DLCTA of the whole aorta using 70 kVp and 200 mg I/kg can provide sufficient CTA attenuation when using an optimal contrast medium delivery protocol for diagnosis and is thus feasible in clinical practice [17]. However, whether DLCTA could reliably detect the presence of endoleak in patients who received TEVAR or EVAR is not clear. Thus, this study aimed to assess the diagnostic performance of DLCTA for aneurysm assessment and endoleak detection compared to the standard CTA (SDCTA) protocol.

## 2. Methods

### 2.1. Study Population

According to HIS and RIS, all patients who received EVAR treatment and underwent serial follow-up CTA from February 2014 to October 2019 were retrospectively reviewed. The inclusion criteria included patients with at least one standard CTA protocol and one DLCTA protocol. The exclusion criteria were poor image quality and incomplete image datasets. In total, 60 patients were enrolled, including 62 standard CTA and 167 DLCTA datasets. The median number of CTAs per patient was five (range: two to eight). The overall diagnosis considering standard CTA and all subsequent CTAs by the consensus of two radiologists (C.P.A. and M.T.W. with 6 and 20 years of experience in cardiovascular radiology) was considered the reference standard for the presence of endoleak (Figure 1).

### 2.2. CT Technique and Acquisition Protocol

All examinations were performed using wide-detector CT scanners (Revolution CT, GE Healthcare, Milwaukee, WI, USA). The following imaging parameters were used: automatic attenuation-based tube current modulation, 0.625 mm detector collimation x 256 rows, pitch value 1, medium-strength iterative reconstruction (ASiR-V 40%), section thickness and intervals 0.625 and 0.625 mm, rotation time 0.35 s.

The 62 standard CTA images were obtained at 120 kVp with 400 mgI/kg of contrast medium (Iohexol, Omnipaque 350 mg iodine/mL; GE Healthcare, Chalfont St. Giles, UK). The 167 DLCTA imaging was obtained at 70 or 80 kVp with 200 mgI/kg of contrast medium. An automatic bolus tracking technique was used to optimize trigger acquisition after injection of the contrast medium. We set an attenuation threshold of 120 HU at the region of interest (ROI) placed over the ascending aorta plus a delay of 5 s before starting the scan. Delay phase acquisition was initiated 60 s after the arterial phase, and the tube voltage was the same as the voltage used in the arterial phase. Our triple-phase protocol includes a non-contrast scan of the whole aorta, arterial phase of the entire aorta, and delay phase of the post-EVAR/TEVAR segments.

### 2.3. Image Reconstruction and Analysis

All images were transferred to a dedicated workstation (GE AW 3.2 workstation) for evaluation. All images were reconstructed with a slice thickness of 2.5 mm in average or maximal intensity projection in multi-planar or curved planar reconstruction. The volume-rendered image was used if needed.

The interpretation process was performed by two radiologists (with 6 and 20 years of experience in radiology) to assess the presence of endoleak on any scan. Initially, the two readers independently read each scan without knowing the identification number of the patients or the findings of previous or following scans. Then, both readers independently read all the serial scans in each case to reach a diagnosis. Finally, for cases with inconsistent findings between the two readers, a consensus reading of the whole scan series was performed for the final diagnosis. All endoleaks were classified as types I to V [19]. The two observers quantified their diagnostic confidence in identifying endoleaks using the following Likert scale: 1 = certain absence, 2 = probable absence, 3 = possible presence, 4 = probable presence, and 5 = certain presence [20]. The image noise and EVAR artifacts were evaluated for the overall subjective image quality assessment using the following Likert scale: 1 = poor, 2 = intermediate, and 3 = good.

### 2.4. Radiation Dose Estimation

The volume CT dose index (CTDI_vol_) and dose–length product (DLP) were recorded and used to calculate the radiation dose. The approximate effective dose (ED) was defined by multiplying the DLP by the conversion factor. We used 0.0145 mSv/mGy·cm, which is the mean of the thoracic (0.014 mSv/mGy·cm) and abdominal (0.015 mSv/mGy·cm) coefficients, for calculation.

### 2.5. Statistical Analysis

SPSS software (version 20 SPSS Inc., Chicago, IL, USA) was used for statistical analysis. To determine the accuracy of endoleak detection, we calculated the sensitivity, specificity, positive predictive value, negative predictive value, and overall accuracy. All numeric values are expressed as means ± SD, and categorical variables are expressed as frequencies or percentages. A two-tailed t-test was used for quantitative variables, including patient characteristics and CTDI_vol_. The chi-square test was used to compare the frequency distribution with regard to gender. A *p*-value of less than 0.05 (*p* < 0.05) indicated statistically significant results.

## 3. Results

### 3.1. Patient Characteristics

There were 7 female and 53 male patients. The mean age was 64.9 ± 11.3 years. The mean BMI was 26.8 ± 3.8. In total, 36 patients received TEVAR, and 24 patients received EVAR. The median number of follow-up CTA scans was five (minimum follow-up: two, maximum follow-up: eight). The mean CTA follow-up interval was 306.7 ± 99.5 days (Table 1). Fifty-one patients received initial SDCTA(s) followed by DLCTA(s). Among these 51 cases, 24 cases received further follow-up SDCTA(s) after the DLCTA(s). Nine patients received initial DLCTA(s) followed by SDCTA(s), and all these nine cases received further follow-up DLCTA(s) after the SDCTA(s).

### 3.2. Endoleak Analysis

We found that 13 (21.7%) patients had endoleak in the first postoperative CTA (9 SDCTA and 4 DLCTA), and 14 (23.3%) patients had endoleak in the subsequent CTA, either DLCTA or SDCTA. The sensitivity, specificity, positive prediction rate, negative prediction rate, and overall accuracy of endoleak detection by DLCTA were 85.7%, 95.7%, 85.7%, 95.7%, and 93.3%, respectively. Thus, the rate of agreement of DLCTA with SDCTA was 93.3% (*n* = 56) and the rate of disagreement was 6.7% (*n* = 4) for the presence of endoleak (Table 2).

Figure 2 demonstrates the case of a patient who underwent TEVAR for thoracic aortic aneurysm, and the DLCTA on delay phase showed a type II endoleak. The size of the endoleak was diminished on the follow-up DLCTA 2 years later.

**Figure 2 diagnostics-13-02228-f002:**
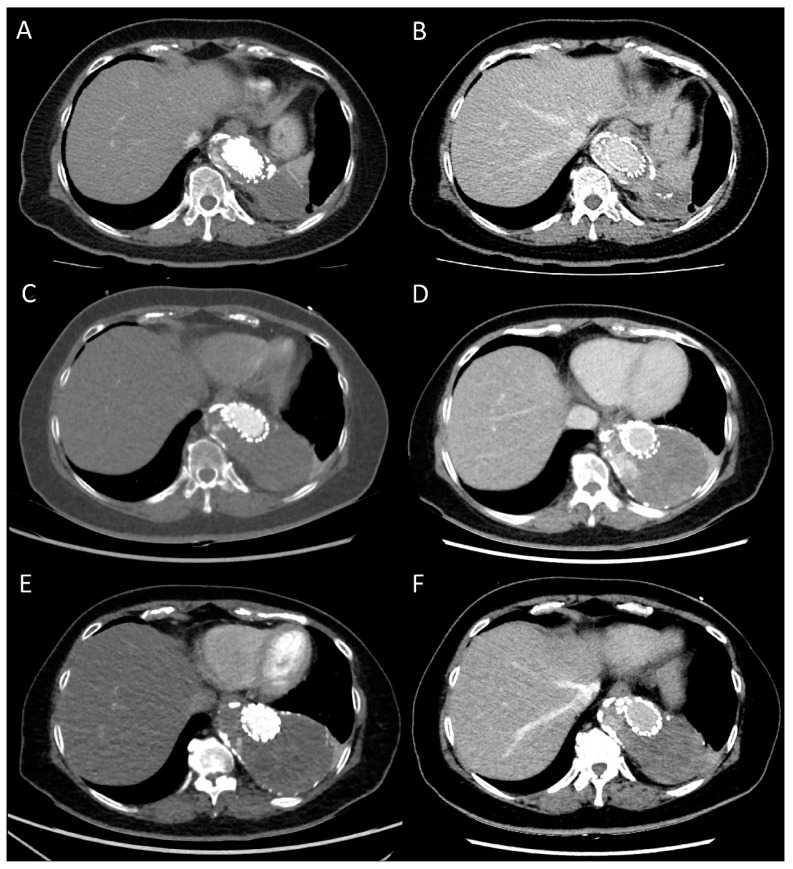
A 63-year-old woman underwent endovascular repair of a thoracic aortic aneurysm (**A**,**B**). The arterial phase (**A**) and delay phase (**B**) of SDCTA scanned at 120 kVp with 70 mL of 350 mg I/mL contrast medium (total iodine dose, 24.5 g) showed type II endoleak within the aneurysmal sac (**C**,**D**). The arterial phase (**C**) and delay phase (**D**) of DLCTA scanned at 70 kVp with 35 mL of 350 mg I/mL contrast medium (total iodine dose, 12.25 g) 2 years after the SDCTA showed persistent type II endoleak within the aneurysmal sac (**E**,**F**). The arterial phase (**E**) and delay phase (**F**) of DLCTA scanned at 80 kVp with 35 mL of 350 mg I/mL contrast medium (total iodine dose, 12.25 g) 2 years after (**C**,**D**) showed type II endoleak within the aneurysmal sac with mild regression. All the images are diagnostic.

**Table 2 diagnostics-13-02228-t002:** Numbers and statistical measures for endoleak detection by standard-dose CTA and DLCTA. The four consensus revision cases are shown in Figure 3, Figure 4 and Figure 5.

**Endoleak Detection by Standard-Dose CTA and DLCTA**								
		SDCTA						
	Endoleak	Positive	Negative	Sensitivity	Specificity	PPV	NPV	Accuracy
DLCTA	Positive	12	2	85.7%	95.7%	85.7%	95.7%	93.3%
	Negative	2	44					
**Endoleak Detection by Standard-Dose CTA and DLCTA (Revised Version)**								
		SDCTA						
	Endoleak	Positive	Negative	Sensitivity	Specificity	PPV	NPV	Sccuracy
DLCTA	Positive	14	0	100%	100%	100%	100%	100%
	Negative	0	46					

Among the 14 cases of endoleak detected from DLCTA, 12 cases showed endoleak in the arterial phase, and 2 cases showed endoleak only in the delay phase. Among the 14 cases of endoleak detected from SDCTA, 12 cases showed endoleak during the arterial phase, and 2 cases showed endoleak only in the delay phase. One and two patients showed more prominent contrast medium leakage in the aneurysm detected in the delay phase compared to the arterial phase by DLCTA and SDCTA, respectively, defined as a certain presence of endoleak (Table 3).

The mean follow-up interval for CTA was 306.7 days. The mean aneurysm sizes were 4.09, 4.01, and 4.03 cm in the initial, midterm, and latest CTA, respectively. In the endoleak group, the mean aneurysm sizes were 4.88, 4.88, and 4.94 cm in the initial, midterm, and latest CTA, respectively. In the non-endoleak group, the mean aneurysm sizes were 3.83, 3.63, and 3.70 cm in the initial, midterm, and latest CTA, respectively (Table 4). There was no significant difference in the aneurysm size in all endoleak and non-endoleak cases. However, four cases revealed aneurysm sac growth of 5 mm or greater during the follow-up CTA.

Four cases showed discordance between the SDCTA and DLCTA. Case 1 had an endoleak in the baseline SDCTA, but no endoleak was found in follow-up DLCTA one year later. We confirmed this as a spontaneous resolution of the endoleak due to a lack of endoleak on follow-up SDCTA after the DLCTA (Figure 3). Case 2 had no endoleak during the initial first-time follow-up DLCTA, but an endoleak was found during follow-up SDCTA one month later. Four months later, an identical endoleak was found on DLCTA. Thus, we concluded that this patient had developed an endoleak, and he received coil embolization for the endoleak.

Case 3 was misdiagnosed as an endoleak on the arterial and delay phases on the initial post-EVAR SDCTA. However, a high-density thrombus was diagnosed on non-contrast SDCT. The follow-up DLCTA showed spontaneous regression of the aneurysm sac size (Figure 4). Case 4 revealed no endoleak in the initial post-EVAR SDCTA or the follow-up SDCTA one year later. However, follow-up DLCTA another year later showed an endoleak. The developing endoleak was confirmed by follow-up SDCTA one year later (Figure 5).

After consensus reading of the whole serial CTA scans of these four patients, we concluded that Case 2 and Case 4 displayed true development of an endoleak, while Case 1 was a true resolution of an endoleak. Case 3 was a misdiagnosis by SDCTA with true regression of the aneurysmal sac. Therefore, we revised the result accordingly. The adjusted sensitivity, specificity, positive prediction rate, negative prediction rate, and overall accuracy of endoleak detection by DLCTA were all 100%.

There were 2 type I (3.3%) and 12 type II endoleaks (20.0%) among the 14 patients with endoleak.

### 3.3. Radiation and Contrast Medium Dose

The mean CTDI_vol_ of the arterial phase of the standard CTA scan and DLCTA scan were 9.94 and 2.85 mGy, respectively. For the whole triple-phase CTA protocol of SDCTA and DLCTA, the radiation doses were 26.2 and 12.7 mSvi, respectively. The mean contrast medium doses used in the SDCTA and DLCTA scans were 396.5 and 199.7 mgI/kg, respectively. Compared to the standard CTA, the DLCTA scan reduced the radiation dose by 71% and the iodine dose by 50%.

## 4. Discussion

DLCTA of the whole aorta at 70–80 kVp with 200 mg I/kg was used in this study to evaluate the aneurysm sac and endoleak detection after EVAR/TEVAR as compared to SDCTA. In our study, the diagnostic performance of DLCTA for endoleak detection was comparable to that of SDCTA. To the best of our knowledge, this is the first study to evaluate the diagnostic accuracy for endoleak detection of DLCTA at 70–80 kVp and with a 50% iodine dose reduction.

Some previous phantom studies showed that CTA at 80 kVp does not increase the risk of overlooking endoleaks. Zsolt et al. showed that CTA at 80 kVp in small and intermediate-sized patients is feasible for detecting endoleaks measuring 6 mm or larger [21]. Deak et al. revealed that endoleaks were detectable in CTA at 80 kVP using an iterative reconstruction algorithm [22].

Several previous studies focused on radiation-dose-saving protocols in post-EVAR CTA. Hansen et al. demonstrated that CTA at 100 kVp with 120 mL of 370 mg I/mL contrast medium and an iterative reconstruction algorithm provides appropriate diagnostic image quality with an average 67.5% dose reduction [23]. Naidu et al. used 100 kVp and 2.2 mL/kg of 350 mg I/mL contrast medium as the parameters for CTA scanning. Endoleaks were detected equally in 5 of the 20 patients in their study by both standard-dose (120 kVp) CTA and low-dose (100 kVp) CTA. [24]. However, the standard iodine dose was used in the abovementioned studies [23,24].

As for the reduction in iodine dose in CTA after EVAR, Buffa et al. claimed that a single-phase CTA with a dual-energy scan during the delayed phase is feasible for endoleak detection. The estimated ED of the delayed dual-energy scan was 10.5 mSv, and the cumulative estimated ED was 27.4 ± 2.6 mSv for the triple-phase acquisition protocol [25]. Patino et al. used dual-energy CTA images (40 and 50 keV) acquired with lower iodine doses (16.0 and 16.2 g) to compare with SDCTA (120 kVp single-energy CTA with a 33 g iodine dose). They showed that the dual-energy CTA could provide appropriate diagnostic image quality with about 29.3% reduced total DLP and about 50% reduced iodine dose compared to SDCTA. They demonstrated that the endoleak detection sensitivity of dual-energy CTA was about 78.9–89.4% and 78.9–94.7% on 40 and 50 keV images, respectively. The specificity was 100% for both 40 and 50 keV images [26]. Compared to their study, our triple-phase DLCTA saved 51.5% of the radiation dose and 50% of the iodine load, with an equal diagnosis accuracy of endoleak detection compared to triple-phase SDCTA.

Due to the retrospective study design, the use of SDCTA and DLCTA was sequentially interchanged. Therefore, integrative reading of the whole CTA series is the key to reaching a consensus about the endoleak’s “true story”, case by case. There were four cases where the detection of endoleak between standard-dose CTA and DLCTA was contradictory in the initial reading session. After reviewing the overall serial CTA scans, we concluded that two of them were resolved endoleaks, another was an evolving endoleak, and the last was an endoleak mimic. Ultimately, there was no discordance between SDCTA and DLCTA in our cases.

Our study has some limitations. First, we did not perform the SDCTA and DLCTA at the same time or within a short period for a head-to-head comparison. Evolution or resolution of endoleaks did occur during the follow-up course and may cause controversial interpretations. Second, the sample size was relatively small. However, the positive rate of endoleak was 13 out of 60, which is close to the reported rate and should be clinically representative [3]. Third, because the given maximal tube current coupled with 70–80 kVp is 500–600 mA, the image quality for patients with high BMI may not be acceptable. Fourth, the diagnosis of the type of endoleak was not confirmed by catheter angiography or other modalities. Fifth, several different types of endografts have been developed for aortic aneurysm, such as polymer-based technologies [27]. However, we did not investigate the relationship between the endovascular graft system and DLCTA.

Based on the results of our study, we assume that DLCTA at 70–80 kVp with 200 mg I/kg can reliably detect the presence of endoleak after EVAR and TEVAR. DLCTA is feasible for long-term surveillance of EVAR and TEVAR, especially for elderly patients with compromised renal function.

## Figures and Tables

**Figure 1 diagnostics-13-02228-f001:**
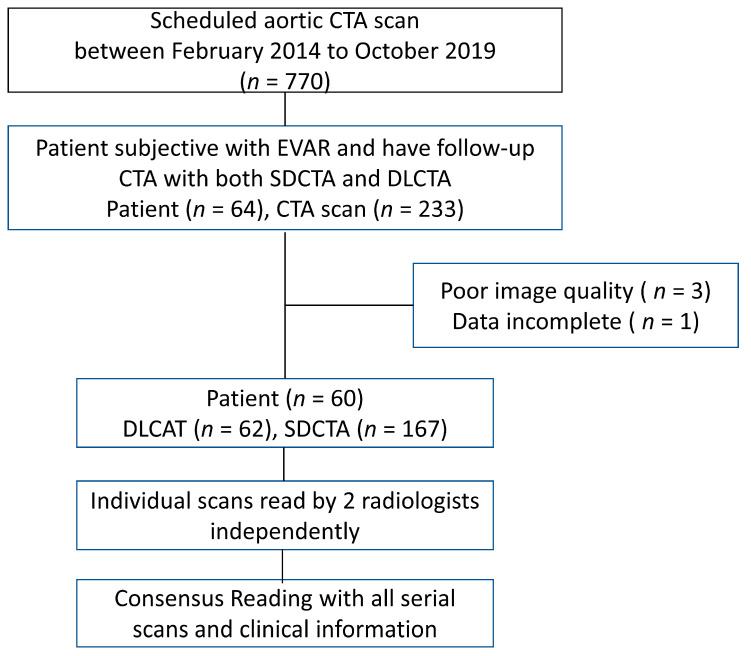
The study design. SDCTA, standard-dose CT angiography; DLCTA, double-low CT angiography.

**Figure 3 diagnostics-13-02228-f003:**
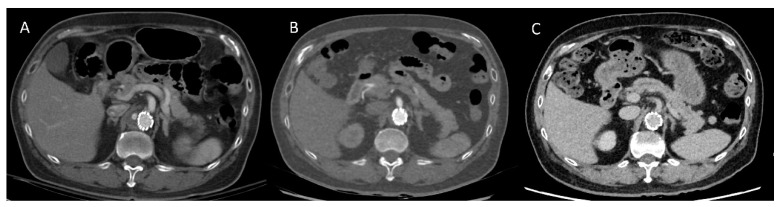
True negative conversion of an endoleak. A 59-year-old man underwent endovascular repair of an abdominal aortic aneurysm. (**A**) The arterial phase of standard-dose CTA scanned at 120 kVp with 90 mL of 350 mg I/mL contrast medium (total iodine dose, 31.5 g) showed type II endoleak within the aneurysmal sac. (**B**) The arterial phase of DLCTA scanned at 70 kVp with 45 mL of 350 mg I/mL contrast medium (total iodine dose, 15.75 g) followed up 2 years later detected no endoleak. (**C**) The delay phase of standard-dose CTA scanned at 100 kVp with 90 mL of 350 mg I/mL contrast medium followed up one year after the CTA shown in (**B**). confirmed resolution of the endoleak and decreased aneurysmal diameter.

**Figure 4 diagnostics-13-02228-f004:**
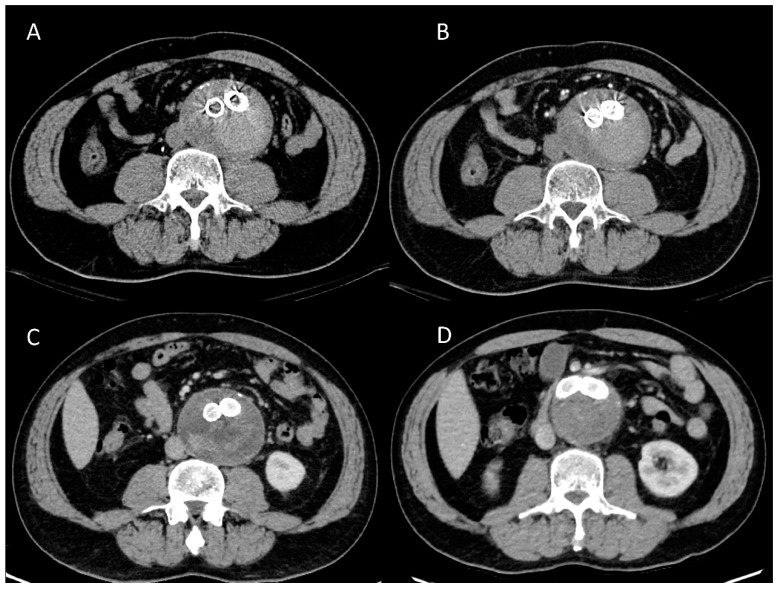
Pseudo-endoleak due to high-density thrombus. A 55-year-old woman underwent endovascular repair of an abdominal aortic aneurysm. Non-contrast CT (**A**) and the arterial phase (**B**) of standard-dose CTA scanned at 120 kVp with 90 mL of 350 mg I/mL contrast medium (total iodine dose, 31.5 g) showed a high-density thrombus within the aneurysmal sac. (**C**) The arterial phase of DLCTA scanned at 70 kVp with 45 mL of 350 mg I/mL contrast medium (total iodine dose, 15.75 g) followed up 1 year later showed heterogenous high density within the aneurysmal sac, which was initially misdiagnosed as type II endoleak. (**D**) The arterial phase of DLCTA scanned at 80 kVp with 45 mL of 350 mg I/mL contrast medium (total iodine dose, 15.75 g) followed up one year after the CTA shown in (**C**). showed decreased aneurysmal diameter and no identifiable high-density lesions, which confirmed the no-endoleak result of the previous DLCTA scan.

**Figure 5 diagnostics-13-02228-f005:**
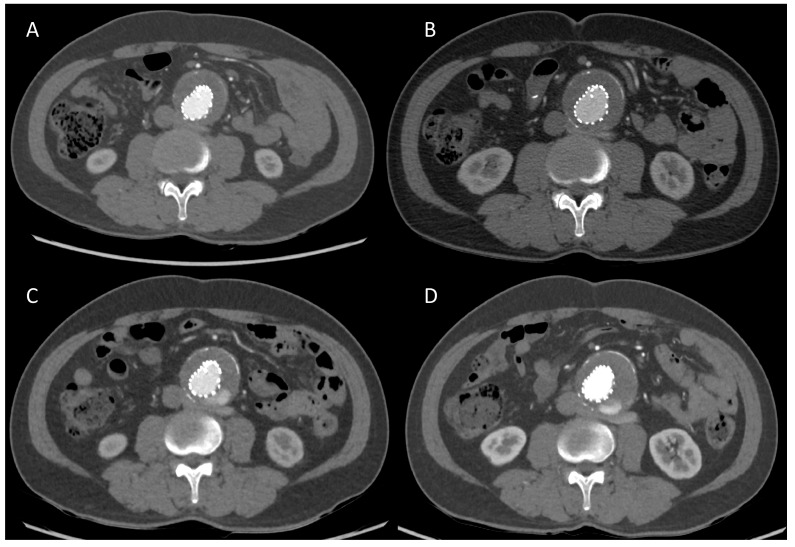
A 66-year-old man underwent endovascular repair of an abdominal aortic aneurysm. (**A**) The arterial phase of standard-dose CTA scanned at 120 kVp with 76 mL of 350 mg I/mL contrast medium detected no endoleak. (**B**) The arterial phase of standard-dose CTA scanned at 120 kVp with 76 mL of 350 mg I/mL contrast medium followed up 1 year after (**A**) revealed no endoleak. (**C**) The arterial phase of DLCTA scanned at 80 kVp with 40 mL of 350 mg I/mL contrast medium followed up 2 years after (**A**) showed type II endoleak within the aneurysmal sac. (**D**) The arterial phase of SDCTA scanned at 100 kVp with 80 mL of 350 mg I/mL contrast medium followed up 3 years after (**A**) also showed type II endoleak within the aneurysmal sac, which confirmed the presence of endoleak in the previous DLCTA scan.

**Table 1 diagnostics-13-02228-t001:** Patient demographics, CTDIvol, and iodine load for double-low CT aortagraphy (DLCTA). TEVAR: thoracic endovascular aneurysm repair; EVAR: endovascular aneurysm repair.

	Value	SD
Total no. of patients	60	
Males	53	
Females	7	
Age	64.9	11.3
BMI	26.8	3.8
Body weight (kg)	74.8	12.9
Body Height (cm)	166.7	7.6
CTDI_vol_ (mGy)	2.85	1.6
Iodine load (mg/kg)	199.7	11.2
Type of stent		
TEVAR	36	
EVAR	24	
Interval of CTA follow-up (days)	306.7	99.5
Number of CTA studies	5.2	1.9

**Table 3 diagnostics-13-02228-t003:** Confidence of endoleak detection and subjective image noise evaluation of standard-dose CTA and DLCTA.

	DLCTA	SDCTA
Arterial-phase endoleak (total)	12	12
Certain absence	2	2
Probable absence	0	0
Possible presence	3	3
Probable presence	9	9
Delay-phase endoleak (total)	14	14
Certain absence	0	0
Probable absence	0	0
Possible presence	1	1
Probable presence	12	11
Certain presence	1	2
Noise level		
Good	11	13
Intermediate	3	1
Poor	0	0

**Table 4 diagnostics-13-02228-t004:** Statistical measures of aneurysm size in serial CTA studies.

	Aneurysm Size (cm)						
	Initial CTA		Midterm CTA		Last CTA		
	Mean	SD	Mean	SD	Mean	SD	*p*
All (*n* = 60)	4.09	0.68	4.01	1.24	4.03	1.02	0.908
Presence of endoleak (*n* = 14)	4.88	0.65	4.88	1.11	4.94	0.99	0.987
Absence of endoleak (*n* = 46)	3.83	0.71	3.63	0.85	3.70	0.88	0.572

## Data Availability

The data that support the findings of this study are available from the corresponding author, M.T.W. upon reasonable request.

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
