# Peer review of "Can Low-Iodine, Low-Radiation-Dose CT Aortogram Reliably Detect Endoleak after Endovascular Aneurysm Repair of the Aorta?"

_diagnostics, 2023, doi:10.3390/diagnostics13132228_

Round 1

Reviewer 1 Report

The Authors presented an interesting paper regarding the use of a dedicated low-iodine and low-radiation CT protocol for EVAR and TEVAR follow-up. 

I think reducing radiation exposure in patients who have to be followed up for their entire life is essential. Also, the reduction of contrast medium exposure is essential. Numerous patients during their life will undergo different types of endovascular procedures with a high risk for chronic renal failure. The implementation of this type of protocol could reduce significant kidney-related injuries, especially in patients with underlying chronic kidney disease. 

The paper is well presented and the results are well reported.

I think that the Authors have to clarify one important thing. The type of endograft that was implanted in this case series. This is essential because different endografts have different radiological features. Especially, the novel platforms that implied polymer-based technology, especially for the detection of Type I EL. How this specific protocol could perform in this type of endografting? Could the Authors comment on this? or exposed this particular topic in the limitation section? [doi: 10.1016/j.ejvsvf.2021.11.003.]

Minor flaws

Reviewer 2 Report

I have read the article "Can low-iodine low-radiation dose CT Aortogram reliably detect the endoleak after endovascular aneurysm repair of the aorta?"

The overall design of the study is retrospective with a relatively small amount of subjects with positive endoleaks; therefore, results could be biased - as already mentioned in the Discussion.

I have found several issues, mainly regarding discrepancy cases:

1) Case 3. There is a hyperdense part of aneurysmal sac content called an atheroma. In fact, this part is adjacent to stent graft and more peripheral parts are more hypodense. Therefore this hyperdense part should be a thrombus and not an atheroma. Therefore it still could represent a small endoleak. Probably pre-procedure CT could be compared if it is true atheroma or thrombus at lumen excluded by stent-graft.

2) Case 4. In this case, there is the same image as case 3 (Figures 4 and 5 are the same)

3) Moreover, based on methodology and all information cannot be written in the abstract that "The performance of DLCTA in the diagnosis of endoleak reached 100% accuracy." As mentioned, some cases are misdiagnosed by standard dose and/or low-dose CT. Therefore sentence should be "The performance of DLCTA in the diagnosis of endoleak could reach 100% accuracy compared to standard CTA in case of correction of CT findings by an expert second reading."

4) One other recent related article should be cited: Fink MA, Stoll S, Melzig C, Steuwe A, Partovi S, Böckler D, Kauczor HU, Rengier F. Prospective Study of Low-Radiation and Low-Iodine Dose Aortic CT Angiography in Obese and Non-Obese Patients: Image Quality and Impact of Patient Characteristics. Diagnostics (Basel). 2022 Mar 10;12(3):675. doi: 10.3390/diagnostics12030675. PMID: 35328228; PMCID: PMC8947155.

After clarifications and corrections, I think the paper is valuable for publishing and could be interesting for readers. 
